# Light Regulates Secreted Metabolite Production and Antagonistic Activity in *Trichoderma*

**DOI:** 10.3390/jof11010009

**Published:** 2024-12-26

**Authors:** Edgardo Ulises Esquivel-Naranjo, Hector Mancilla-Diaz, Rudi Marquez-Mazlin, Hossein Alizadeh, Diwakar Kandula, John Hampton, Artemio Mendoza-Mendoza

**Affiliations:** 1Faculty of Agriculture and Life Sciences, Lincoln University, Lincoln 7647, New Zealand; hossein.alizadeh@lincoln.ac.nz (H.A.); diwakar.kandula@lincoln.ac.nz (D.K.); 2Unit for Basic and Applied Microbiology, Faculty of Natural Sciences, Autonomous University of Queretaro, Queretaro 76230, Mexico; 3School of Physical and Chemical Sciences, University of Canterbury, Christchurch 8041, New Zealand; hector.mancilladiaz@canterbury.ac.nz (H.M.-D.); rudi.marquez-mazlin@canterbury.ac.nz (R.M.-M.)

**Keywords:** crop protection, plant pathogen, biocontrol, antagonism, secondary metabolism, fungal biotechnology

## Abstract

Secondary metabolism is one of the main mechanisms *Trichoderma* uses to explore and colonize new niches, and 6-pentyl-α-pyrone (6-PP) is an important secondary metabolite in this process. This work focused on standardizing a method to investigate the production of 6-PP. Ethanol and ethyl acetate were both effective solvents for quantifying 6-PP in solution and had limited solubility in potato–dextrose–broth media. The 6-PP extraction using ethyl acetate provided a rapid and efficient process to recover this metabolite. The 6-PP was readily produced during the development of *Trichoderma atroviride* growing in the dark, but light suppressed its production. The 6-PP was purified, and its spectrum by nuclear magnetic resonance and mass spectroscopy was identical to that of commercial 6-PP. Light also induced or suppressed other unidentified metabolites in several other species of *Trichoderma*. The antagonistic activity of *T. atroviride* was influenced by light, as suppression of plant pathogens was greater in the dark. The secreted metabolite production on potato–dextrose–agar was differentially regulated by light, indicating that *Trichoderma* produced several metabolites with antagonistic activity against plant pathogens. Light has an important influence on the secondary metabolism and antagonistic activity of *Trichoderma*, and this trait is of key relevance for selecting antagonistic *Trichoderma* strains for plant protection.

## 1. Introduction

Several species of the genus *Trichoderma* are cosmopolitan fungi commonly found in soil. Their association with plants can result in the promotion of plant growth and an increased ability of the plant to cope with abiotic and biotic stress [1]. Many *Trichoderma* species can protect crops against plant pathogens [1,2] using mechanisms such as competition for space and nutrients, antibiosis, mycoparasitism, and indirectly promoting pathogen–defense mechanisms in plants [1,2]. Mycoparasitism has been proposed as the ancestral life of the genus *Trichoderma* [3] and involves the sequential stages of recognition of the plant pathogen, hydrolytic enzyme production, destroying the cell wall, coiling around the host hypha, penetration, and growing inside the host hyphae [2,4]. Antibiosis strengthens the ability of *Trichoderma* to destroy fungal plant pathogens by a synergistic action with hydrolytic enzymes. This confers a competitive advantage for the colonization of plant pathogens as a mycoparasite, different substrates such as organic material like saprobe, and/or the rhizosphere such as an endophyte [1,2,5].

Available genome sequences show that *Trichoderma* spp. have a large genetic repository to produce many secondary metabolites, including non-ribosomal peptides, polyketides, terpenoids, and phytohormones [5,6]. *Trichoderma* can also produce volatile and non-volatile organic compounds such as polyphenols, flavonoids, terpenes, pyrones, sesquiterpenes, ketones, thioesters, cyclohexanes, and alcohols [7,8,9,10]. Some of these compounds, which are available commercially, have been assessed for their potential for crop protection against plant pathogens [8,9,10,11,12] and/or their ability to function as phytohormones, regulating root development, leaf area, photosynthesis, and promoting plant growth [1,10,13,14]. Additionally, *Trichoderma* secrete siderophores and organic acids, increasing soil nutrient availability for plant uptake [1,5,15].

One of the main volatile organic compounds emitted by several *Trichoderma* species is 6-pentyl-α-pyrone (6-PP) [16]. It has antagonistic activity against many plant pathogens [8,11,12] but also has phytohormone-like activity via regulating root development and plant growth and inducing systemic resistance [10,16,17,18]. While its biosynthesis pathway is still unknown, the production of 6-PP is induced during the interaction between fungi and plants [16,19]. Intricate regulation through the mitogen-activated protein kinase (MAPK) and cyclic adenosine monophosphate (cAMP) via signaling pathways, enzymes producing reactive oxygen species, and light photoreceptors have been reported to regulate 6-PP production from *Trichoderma* colonies [8,16,20,21]. Different light wavelengths can suppress its production [22]. Plant pathogens overgrew *Trichoderma* lacking in the MAPK Tmk3 in light [22], suggesting that light, through this MAPK, regulates *Trichoderma’s* capacity to colonize plant pathogens. Furthermore, there is evidence of a dual role for 6-PP, which can act either at short distances as a diffusible factor or at long distances as a volatile [9,23].

The method used to investigate the volatiles, including 6-PP produced by *Trichoderma*, uses closed chambers coupled to resins that capture the volatiles. They are then identified by gas chromatography (GC) coupled to mass spectroscopy (MS) [9,11]. However, *Trichoderma* can produce 6-PP in submerged cultures [12] and also probably other volatile and non-volatile organic compounds. From this perspective, *Trichoderma* growing in submerged cultures offers advantages for directly analyzing secondary metabolite production by extraction using solvents and for determining their functional role during the interactions with biotic and abiotic factors.

The main aim of our research was to (i) standardize the method for extraction of secreted secondary metabolites, (ii) investigate the impact of light on their production, and (iii) determine how this regulation by light impacted the antagonistic activity of *T. atroviride* against several fungal plant pathogens.

## 2. Materials and Methods

### 2.1. Trichoderma Strains and Plant Pathogens

The *Trichoderma* strains used in this work were *Trichoderma atroviride* IMI206040 (ATCC 204676), *Trichoderma* sp. *atroviride* B LU132, *Trichoderma* sp. *atroviride* B LU584, *Trichoderma* sp. *atroviride* B LU633, *T. hamatum* LU592, *T. asperellum* LU697, *T. gamsii* LU755, and *T. viridescens* LU1369. The strains with LU codes were New Zealand isolates [24] obtained from the Lincoln University *Trichoderma* collection. Isolates were cultured for seven days at 27 °C on potato–dextrose–agar (PDA, DIFCO, Sparks, MD, USA) in a growth chamber equipped with white light (112 µmoles/m^2^s) to produce conidia and in liquid medium potato–dextrose broth (PDB, DIFCO, Sparks, MD, USA) using a shaker inside the same growth chamber. Conidia were harvested by scraping the surface of the plates with a hockey stick, added to sterile water to make a suspension, counted using a Neubauer chamber, and kept at 4 °C. The liquid medium cultures were used to analyze the metabolites secreted by *Trichoderma*.

For the experiment to assess the antagonistic activity of *Trichoderma* spp., the plant pathogens selected were pathogens of above-ground plant tissues: *Alternaria alternata* [25] and *Botrytis cinerea* [26]; roots: *Gaeumannomyces graminis* var *tritici* [27], *Rhizoctonia solani* [28], *Sclerotinia sclerotiorum* [29], and *Phytophthora cinnamomi* (Ph478 and Ph3795 strains) [30]; or both: *Colletotrichum graminicola* ICMP12090 [31], and *Fusarium oxysporum* [32]. The pathogens were cultured for 7 days at 27 °C on PDA (or V8 medium for the oomycetes), except for *B. cinerea,* which was grown at 20 °C.

### 2.2. UV–VIS Analysis of 6-Pentyl-α-pyrone in Different Solvents

The 6-PP (Sigma-Aldrich, St. Louis, MO, USA) was dissolved in solvents with different relative polarities (RP): hexane (RP = 0.006, MERK, Darmstadt, Germany); ethyl acetate (RP = 0.228, HPLC grade, Fisher Chemical, Geel, Belgium); ethanol (RP = 0.654, absolute AR grade, Fontenay-sous-Bois France); and PDB dissolved in water (RP = 1) [33] to determine the wavelength absorbed by 6-PP. A spectrum scan was carried out from 200 to 700 nm using a UV-1600PC spectrophotometer (VWR International Europe bvba Researchpark Haasrode 2020, Geldenaaksebaan 464, B 3001 Leuven, Belgium, M. Wave Professional 1.0) to identify the maximum absorbance peak at different wavelengths. All experiments were repeated four times.

### 2.3. Dynamic Linear Range to Quantify 6-Pentyl-α-pyrone

To quantify 6-PP, different concentrations of 6-PP were dissolved in hexane, ethanol, ethyl acetate, and PDB. First, a solution of 10 g/L of 6-PP was prepared and then used as a stock solution to prepare the less concentrated solutions (1, 2.5, 5, 10, 20, 40, 60, 80, 100 mg/L) to identify the concentrations where the 6-PP linearly absorbed light. The concentrations between linear ranges were then analyzed around the absorbance maximum peak at different wavelengths to make a standard curve.

### 2.4. Extraction of 6-Pentyl-α-pyrone from PDB

Ethyl acetate was used to recover 6-PP from the media because the solubility of 6-PP is low in water. Briefly, a solution of 50 mg/L of 6-PP was prepared in PDB; the medium containing 6-PP was mixed with ethyl acetate (1:1 *v*/*v*), vortexed at 3400 rpm (Labnet Vortex Mixer, International Labnet, Cary, NC USA) for different times (15 and 30 s; 1, 2, 5, 10, and 15 min), and finally centrifuged for 3 min at 3000 rpm (Eppendorf centrifuge 5810 R, Hamburg, NY, USA) to separate the organic phase from the media. The supernatant was recovered, and 6-PP was quantified using the standard curve of 6-PP dissolved in ethyl acetate. This approach allowed for the determination of the optimal mixing time.

To investigate the possibility of detection of a higher concentration of the metabolites recovered directly from the media, a solution of 20 mg/L of 6-PP was prepared in PDB and mixed at the following volume of solvent(S): media(M) ratios (1S:1M, 1S:2M, 1S:3M, and 1S:4M). These were then vortexed at 3400 rpm for 30 s, and the mixture was centrifuged for 3 min at 3000 rpm. The 6-PP was quantified from the supernatant recovered using the standard curve of 6-PP dissolved in ethyl acetate.

### 2.5. Extraction of Metabolites from Liquid Media

*Trichoderma atroviride* IMI206040 (1 × 10^6^ conidia per mL) was added to 125 mL flasks containing 25 mL of PDB and incubated in constant dark, constant white light, or under a 12 h dark:12 h light photoperiod at 27 °C for 7 days on a shaker at 120 rpm. A time course experiment was conducted with 25 mL cultures sampled every 24 h up to 168 h (7 days). Three biological replicates were filtered independently using Miracloth tissue (Merck Millipore, Auckland, New Zealand) at each time point. The filtrates were placed into plastic falcon tubes for metabolite analysis, and the Miracloth containing the mycelia was dried at 80 °C for 24 h to determine mycelium dry weight. The metabolites were extracted by mixing 3 mL of filtrate with 3 mL of ethyl acetate, vortexed for 1 min at maximum speed (3400 rpm), and then centrifuged for 3 min at 3000 rpm. The resultant supernatant was used to analyze the absorbance spectrum using a UV-1600PC spectrophotometer (VWR), and thin layer chromatography (TLC) was performed by mixing 2 mL of each replicate to obtain 6 mL of filtrate and 1.5 mL of ethyl acetate.

Metabolites from the different *Trichoderma* strains (IMI206040, LU132, LU584, LU633, LU592, LU697, LU755, LU1369) were compared by inoculating 1 × 10^6^ conidia per mL of each strain into the flasks containing 25 mL of PDB and incubated in the dark or light for 3 days at 27 °C.

### 2.6. Thin Layer Chromatography Assay

Metabolites were separated using thin-layer chromatography (TLC). For submerged cultures, 1.5 mL ethyl acetate and 6 mL of filtrate were mixed, vortexed for 1 min at maximum speed (3400 rpm), and centrifuged for 3 min at 3000 rpm. The supernatant was recovered and kept at −20 °C. Aluminium TLC plates (20 × 20 cm) covered with silica gel (60 F_245_, MERCK) were used for metabolite separation. Ten 5 µL drops (50 µL per sample) of each sample were applied, and the TLC left to dry before the TLC plates were carefully put into the glass chamber containing 50 mL of the mobile phase *n*-hexane:diethyl ether:acetic acid (70:30:1). Before placing the plates, the chamber had previously been placed inside a fume hood and saturated for 20 min. The metabolites were separated for an hour. The spots were visualized by irradiating the TLC plates both with short-wave (254 nm) and long-wave (356 nm) UV using a mineralight lamp (Model UVGL-58, Multiband UV 254/365 nm). The commercial 6-PP from Sigma-Aldrich was used as a reference control.

### 2.7. Isolation and Characterization of 6-Pentyl-α-pyrone from the Ethyl Acetate Extract

*T. atroviride* IMI206040 metabolites from 1 L of the filtrate were extracted with ethyl acetate and evaporated to 30 mL. The resulting mixture was placed in a separating funnel, and the organic layer was recovered and washed with brine (30 mL). The organic phase was collected in a 100 mL beaker, and 5 g of sodium sulfate was added and allowed to stand for 10 min before being filtered through a cotton plug and concentrated in vacuo. The extract containing the metabolites was then purified by flash chromatography (Silica gel, elution gradient 80:20 (*n*-Hexane: Ethyl acetate) to 70:30 (*n*-Hexane: Ethyl acetate), to yield two products: product A (5.2 mg) and product B (0.5 mg).

Product A was then characterized through proton magnetic resonance spectra (^1^H NMR) and carbon magnetic resonance spectra (^13^C NMR) recorded at 400 MHz and 101 MHz using a JEOL JNM-ECZ400S 400 MHz spectrometer (JEOL, Peabody, MA, USA) NMR with an ASC64 Autosampler. High-resolution accurate mass samples were analyzed on a Synapt XS (Waters Corporation, Milford, MA, USA) coupled with an Acquity Premier Ultra Performance Liquid Chromatography (UPLC) (Waters Corporation, Milford, MA, USA). The sample (0.5 uL) was injected into a flow of 50:50 water (0.5% formic acid):acetonitrile at 0.2 mL/min. Leucine Enkephalin was injected as a Lock Mass. The data were acquired using MassLynx software (Waters Corporation, Milford, MA, USA) and processed using Waters_Connect and UNIFI.

### 2.8. Antagonistic Activity Under Light and Dark Conditions

A cellophane assay was chosen to analyze the impact of light on *Trichoderma* antagonistic activity against the pathogens. Cultures of *Trichoderma* strains (IMI206040, LU584, LU592, LU697, LU755, and LU1369) were grown by inoculating 2 µL of conidia on PDA plates and incubating at 27 °C for 48 h. Mycelia discs 5 mm in diameter were cut from the colony periphery, inoculated on fresh PDA plates covered with a cellophane sheet, and incubated for 48 h at 27 °C under either dark or white light. Then, the cellophane with *Trichoderma* colonies was removed, and a pathogen was inoculated using a mycelial disc cut from the colony margin. These were two days old for *B. cinerea*, *R. solani*, and *S. sclerotiorum* and four days old for the other pathogens. As controls, the pathogens were also inoculated on fresh PDA plates. The colonies were photographed when the control had almost reached the border of the plates: two days for *S. sclerotiorum* and *B. cinerea*, three days for *R. solani*, five days for *A. alternata*, *F. oxysporum,* and *G. graminis*, seven days for *P. cinnamomi*, and 10 days for *C. graminicola*, and diameter of each colony was measured across five directions using ImageJ (Version 1.53). The assays were repeated three times.

Antagonistic activity was determined from the colony diameters of the plant pathogens grown on PDA plates with metabolites produced by *Trichoderma* in the dark and light and compared with the colony diameter on fresh PDA as a control. The inhibition percentage (IP) was determined using the equation IP = 100 × (1 − [CT/CU]), where CT represented the colony treated with metabolites delivery through the cellophane in the dark or light, and CU represented the colony untreated, which was grown on fresh PDA as a control. Average data of the results performed in triplicate were plotted using a matrix heatmap online (https://www.bioinformatics.com.cn/plot_basic_matrix_heatmap_064_en, accessed on 15 October 2024).

### 2.9. Extraction of Metabolites from PDA

Fresh PDA plates covered with a cellophane sheet were inoculated with *Trichoderma* strains (IMI206040, LU584, LU592, LU697, LU755, and LU1369; three plates per strain) using 5 mm mycelial discs from the colony margin. The plates (9 cm diameter) were incubated for 60 h at 27 °C under either constant dark or white light. Then, the cellophane with *Trichoderma* colonies was removed, and the agar was cut into small fragments with a scalpel. All the fragmented agar was put inside 50 mL plastic tubes containing 10 mL ethyl acetate homogenized for 1 min and centrifuged for 3 min at 3000 rpm to separate the supernatant. A 3 mL supernatant was concentrated to 100 µL, and then, 5 µL was used to separate the compounds by TLC. The compounds were separated, as described in Section 2.6.

### 2.10. Statistical Analysis

Microsoft Excel worksheet was used to determine averages and standard deviations of 6-PP concentrations and standard curves, growth rate from dry weights, inhibition growth from the diameters of colonies, and graph generation. All treatments had three replicates, and R was used for statistical analysis [34]. Significant differences were determined using Tukey and Two-way ANOVA tests with a value of *p* ≤ 0.05.

## 3. Results

### 3.1. Dynamic Linear Range to Quantify 6-Pentyl-α-pyrone

The absorption spectrum of 6-PP at 100 mg/L did not differ for *n*-hexane, ethanol, and ethyl acetate. The maximum absorbance reached was 2.6, with a defined peak at 312 nm (Figure 1A). However, in PDB, the maximum absorbance reached was 1.6, with a peak at 323 nm. For the 6-PP concentrations dissolved in *n*-hexane, ethanol, and ethyl acetate, the wavelength range was 301–313 nm, whereas in PDB, the peak range was 311–323 nm (Figure 1A and Appendix A).

For *n*-hexane, the wavelength values were widely dispersed among the 6-PP concentrations assayed (Appendix A), while there was no linear range using PDB as a solvent (Appendix A). At 320 nm, 6-PP had a dynamic linear range of 1–100 mg/L when ethanol or ethyl acetate were the solvents (Appendix A). There was a strong correlation (R^2^) between 6-PP concentration and absorbance for both ethanol (Figure 1B) and ethyl acetate (Figure 1C).

### 3.2. Extraction of 6-Pentyl-α-pyrone from PDB

For all mixing times, 6-PP was rapidly detected in the supernatant, and the shortest mixing time (15 s) was sufficient to recover all 6-PP dissolved in PDB (Figure 2A). This indicates that ethyl acetate is highly efficient in dissolving this compound and facilitating 6-PP recovery from the media. With this solvent’s very low solubility in water, the supernatant was separated after 10 min without any agitation.

When the volume of the PDB media was increased, but the solvent was kept constant, the 6-PP recovered was almost proportional to the media volume (Figure 2B).

### 3.3. Light Regulates Growth and 6-Pentyl-α-pyrone Production in T. atroviride

Given that 6-PP production by *T. atroviride* is regulated by light [22], ethyl acetate extraction was used to analyze 6-PP production by *T. atroviride* IMI206040 growing for 7 days (sampled every 24 h) in PDB under light and dark. Light reduced *T. atroviride* IMI206040 growth, with more mycelium being produced in the dark (Figure 3A). The absorbance spectrum also differed between dark and light (Figure 3B). The peak started increasing at 280 nm, with a maximum at around 300 nm, following which the absorbance dropped (Appendix A). From day three, the absorbance was always around 4.5 times higher in the dark than in the light (Figure 3B). The absorbance reached at 3 days in the dark was very similar to that for days four to seven (Appendix A), suggesting that metabolites absorbing light had reached a maximum level at three days, and after that, only minor changes occurred.

Using TLC, two compounds extractable by ethyl acetate were detected in the dark, Dark Compound 1 (DC1, Rf = 0.21) and another one with the same mobility (Rf = 0.28) as the commercial 6-PP used as a reference, and in the light, at least one compound (Light compound 1, LC1) with a Rf = 0.45 was detected (Figure 3C,D). The compound with less mobility (DC1) was more polar, while LC1, which had more mobility, was less polar than the commercial 6-PP. The compound with the same mobility as 6-PP was partially detected with both short (254 nm) and long (356 nm) UV wavelengths, but the 6-PP from Sigma was clearly detected (Figure 3C,D), suggesting that 6-PP was produced at a low concentration under these experimental conditions. Both DC1 and LC1 were clear in both short and long wavelengths, suggesting that they could absorb a wider spectrum of light, although they could possibly be produced in higher concentrations. After four days, DC1 and LC1 concentrations decreased, whereas the 6-PP-like compound showed a small increase (Figure 3C,D). These results indicate that light inhibited the growth of *T. atroviride* IMI206040 and its production of different secreted organic compounds.

When cultured in a photoperiod of 12 h dark and 12 h light, the growth of *T. atroviride* IMI206040 was similar to that in the dark for the first two days but then reduced (Appendix A). The absorbance was initially higher at 2 days in the dark but thereafter did not differ (Appendix A). The compounds detected were similarly regulated. DC1 and a 6-PP-like compound were at higher levels in the dark, whereas LC1 was higher in the 12 h dark/12 h light treatments (Appendix A). The highest concentrations were detected on the second and third day of incubation, but after four days, the concentrations decreased. These results indicate that growth and metabolite production were regulated by light in *T. atroviride* IMI206040 [35].

### 3.4. Isolation and Characterization of 6-PP from the Ethyl Acetate Extract

When the extracts of *T. atroviride* grown in the dark were analyzed by TLC, a metabolite with the same retention factor as 6-PP was observed. To determine the molecular structure of this metabolite (Figure 4A), we first isolated the compound from the mixture of metabolites using a flash chromatography column. To confirm the structure of the isolated compound ^1^H NMR and ^13^C NMR, these experiments were conducted [^1^H NMR (CD_3_OD, 600 MHz) δ: 7.46 (1H, t, *J* = 9.0, 6.6 Hz, ArC_(3)_**H**), 6.20 (1H, dq, *J* = 6.6, 1.2 Hz, ArC_(2)_**H**), 6.16 (1H, dq, *J* = 9.6, 0.6 Hz, ArC_(4)_**H**), 2.53 (1H, t, *J* = 7.8 Hz, C_(6)_**H_2_**), 1.67 (2H, quint, *J* = 7.2 Hz, C_(7)_**H_6_**), 1.39–1.34 (4H, m, C_(8)_**H_2_
**+ C_(9)_**H_2_**), 0.91 (3H, t, *J* = 7.2 Hz, C_(10)_**H_3_**). ^13^C NMR (CD_3_OD, 151 MHz) δ 168.1 (**C**_(1)_), 165.2 (**C**_(5)_), 146.5 (**C**_(3)_), 113.4 (**C**_(2)_), 104.5 (**C**_(4)_), 34.4 (**C**_(6)_), 32.2 (**C**_(7)_), 27.7 (**C**_(8)_), 23.3 (**C**_(9)_), 14.2 (**C**_(10)_).] (Figure 4A,B).

A direct comparison of the isolated compound’s ^1^H NMR spectrum with that of commercially available 6-PP, as shown in Figure 4B, strongly suggests that the metabolite produced has the same molecular structure as 6-PP. The HRMS further supports this conclusion, showing a molecular weight corresponding to 6-PP plus a proton [HRMS (ESI) calculated for C_10_H_14_O_2_ [M + H]+: *m*/*z* 167.1072, found *m*/*z* 167.1064], demonstrating that 6-PP was produced in submerged cultures of *T. atroviride* IMI206040.

### 3.5. Metabolite Production Is Regulated by Light in Submerged Cultures of Trichoderma spp.

The response to light differed among the *Trichoderma* species/strains (Figure 5A). Light restricted the growth of *T. atroviride* IMI206040 and *Trichoderma* sp. *atroviride* B strains LU132, LU584, and LU633 but did not affect the growth of *T. hamatum* LU592, *T. asperellum* LU697, *T. gamsii* LU755, and *T. viridescens* LU1369 (Figure 5A) in submerged cultures.

The metabolites secreted were also affected by light, with LC1 induced and 6-PP repressed, as previously demonstrated. The *Trichoderma* sp. *atroviride* B produced the metabolites LC1 and LC2 induced by the light, but DC2, DC3, DC4, and 6-PP were repressed by light, a response similar to *T. atroviride* IMI206040 (Figure 5B). Interestingly, the strains of *Trichoderma,* which were not affected by light, produced 6-PP at comparable levels in both dark and light, but other metabolites were regulated by light. This suggests that *Trichoderma* strains are responsive to light, but their growth and 6-PP regulation differ. *T. hamatum* LU592 produced DC2 and DC3 in the dark and LC2 in the light. *T. asperellum* LU697 produced LC2 in the light, and *T. gamsii* produced DC1 and DC3 in the dark and LC2 in the light. *T. gamsii* LU755 produced the strongest signal with the same mobility as 6-PP, both in the dark and in light. *T. viridescens* LU1369 only produced a low concentration of metabolites, and their regulation by light was difficult to evaluate. Metabolite patterns among the different *Trichoderma* strains were regulated by light, but there were variations in metabolite production and their regulation among them (Figure 5B).

### 3.6. Light Influences the Antagonistic Capacity of Trichoderma spp.

*T. atroviride* IMI206040 significantly reduced the growth of all the plant pathogens (Figure 6), although antagonistic activity was greater against the pathogens that infected plant roots (*G. graminis*, *R. solani*, *S. sclerotiorum*, and *P. cinnamomi*) than against the pathogens mainly infecting aerial parts of the plants (*A. alternata*, *B. cinerea*, *C. graminicola*, and *F. oxysporum*). The antagonistic activity was mostly greater in the dark, with 100% inhibition of the growth of *G. graminis*, *S. sclerotiorum*, and *P. cinnamomi* and 60% inhibition of the growth of *A. alternata* (Figure 6A,B). In the light, *T. atroviride* activity against *A. alternata*, *G. graminis*, *S. sclerotiorum*, and *P. cinnamomi* was reduced, but it did not differ from the responses in the dark for *B. cinerea*, *F. oxysporum,* and *R. solani* (Figure 6B).

Like *T. atroviride* IMI206040, light affected the antagonistic activity of *T. atroviride* LU584 and *T. viridescens* LU1369 (Figure 7 and Appendix A). Surprisingly, light did not affect the antagonistic activity of *T. hamatum* LU592 against *S. sclerotiorum,* as inhibition was strong in both conditions. Similarly, in *T. gamsii* LU755, light did not affect antagonistic activity against *G. graminis*. Interestingly, the antagonistic activity of *T. asperellum* LU697 against *A. alternata*, *G. graminis*, *S. sclerotiorum,* and *P. cinnamomi* was not affected by light, and it was as strong in the light as in the dark (Figure 7). These results suggest that *Trichoderma* produces different compounds to antagonize plant pathogens, or their regulation can differ.

The antagonistic activity was higher in *Trichoderma* strains growing in the dark, with the clustering indicating *T. atroviride* IMI206040, *T. hamatum* LU592, and *T. gamsii* LU755 as the most antagonistic strains against the four pathogens analyzed (Figure 7, Appendix A). The effect of metabolites produced by *T. asperellum* LU697 in the dark was as effective as in light, and both treatments were grouped (Figure 7), indicating that light did not affect the secreted metabolite production with antagonistic activity against plant pathogens (Appendix A). The plant pathogens more susceptible to metabolites produced by *Trichoderma* spp. were *S. sclerotiorum*, *G. graminis,* and *P. cinnamomi*.

### 3.7. Light Regulates Secreted Metabolites in Trichoderma spp. Growing on PDA

To analyze the relation between secreted metabolites and antagonistic activity regulated by light, secreted metabolites produced by *Trichoderma* growing on PDA plates were analyzed. The two main compounds produced in the dark by *T. atroviride* IMI206040 and *Trichoderma* sp. *atroviride* B LU584 in submerged culture were also detected on PDA: 6-PP and the compound DC1. The DC1 concentration was decreased in both strains grown in light, while 6-PP decreased only in *T. atroviride* IMI206040, *T. hamatum* LU592, *T. asperellum* LU697, *T. gamsii* LU755, and *T. viridescens* LU1369, but the light did not affect LU584 (Figure 8). Notably, 6-PP was not detected in *T. hamatum* LU592 growing on PDA, whereas in PDB, it was produced, suggesting that in addition to 6-PP, *Trichoderma* produced other compounds with antifungal activity. Although the antagonistic activity of *T. asperellum* LU697 was not affected by light, the secreted metabolite production was regulated (Figure 8). Overall, light regulates secondary metabolism, which, in general, is related to the antagonistic activity of *Trichoderma* spp.

## 4. Discussion

Species of *Trichoderma* commonly colonize the rhizosphere, and some are antagonists of plant pathogens. Secondary metabolism may play an important role in antagonizing pathogens and establishing a beneficial communication dialogue with the plant [1,16]. The 6-PP, the main volatile organic compound secreted by several species of *Trichoderma*, has the properties that *Trichoderma* needs to compete and colonize the rhizosphere [16].

The quantification of 6-PP has been reported using HPLC with a diode-array UV–VIS detector [36]; however, more simple methods for quantifying 6-PP from solution offer routine application advantages, such as standardizing 6-PP and metabolites extraction to analyze their regulation. Solvents such as ethyl acetate or ethanol readily allowed for the detection of 6-PP by absorbance. The dynamic linear range recorded was the same for both solvents. The slope in the equation was very similar (0.0253 and 0.0251, respectively), even though they had different relative polarities. The extraction of secondary metabolites from cultures of *Trichoderma* using organic solvents has been used as a routine method, but the process often involves using large volumes of solvent, iterative extractions, or a long extraction time [37,38]. As the linear relationships found between absorbance and 6-PP concentration when using ethyl acetate as the solvent are significant (Figure 1C), this method is proposed as the standard for quantifying 6-PP. With its limited solubility in water, 6-PP can be extracted very rapidly by this method. Including a centrifuge step in the method may also be unnecessary as the supernatant is readily separated from the non-agitated extraction mixture.

Light regulates growth, metabolism, stress responses, and asexual reproduction in *T. atroviride* through photoreceptors and the MAPK Tmk3 pathway [22,35]. On PDA plates, light suppressed 6-PP production through the MAPK Tmk3 [22]. Although *Trichoderma* produces 6-PP in submerged cultures, its regulation by light has not been previously analyzed. *T. atroviride* grows more slowly in the light than in the dark on PDA [35,39], but in submerged culture, growth inhibition was pronounced at around 35%. In *T. atroviride* IMI206040, the absorbance in the filtrates from two days of growth did not differ between light and dark, but after that, it was higher in the dark than in the light. Consistently, light suppressed the metabolites DC1 and 6-PP when *T. atroviride* was grown in PDB or on PDA, and LC1 was induced in PDB, but it was not detected in PDA, indicating some differences in the regulation of secreted metabolite production. In the dark, a compound of unknown structure (DC1) and a metabolite with the same Rf as 6-PP was observed. To confirm if this metabolite was indeed 6-PP, the product was isolated using flash column chromatography (FCC) and characterized chemically using NMR and HRMS. The spectral data of the isolated metabolite matched that of the reference 6-PP, indicating that it was the same compound previously identified as the primary volatile produced by several species of *Trichoderma* [16].

Once this metabolite was identified as 6-PP, several species of *Trichoderma* were assessed for the effect of light on their production of secondary metabolites in submerged cultures. The strains LU132, LU584, and LU633 of *Trichoderma* sp. *atroviride B* and *T. atroviride* IMI206040 grew more slowly under constant white light than in the dark, whereas the growth of *T. hamatum* LU592, *T. asperellum* LU697, *T. gamsii* LU755, and *T. viridescens* LU1369 did not differ between the conditions, suggesting a difference in light regulation. All the *Trichoderma* species produced a compound that had the same Rf as the commercial 6-PP, and all of them have previously been reported to produce 6-PP [16]. This approach detected two compounds (LC1 and LC2) induced by light and five compounds (6-PP and DC1-4) suppressed by light. LC1 was only produced by *T. atroviride* IMI206040, but LC2 was produced by all the species analyzed. The TLC results suggested that some species produced 6-PP equally well in the dark and light and that *T. gamsii* produced high levels of 6-PP. These results shed light on the significant impact of light on the secondary metabolism of *Trichoderma*, providing evidence of the impact of light on the regulation of secondary metabolism, metabolite production patterns, and divergence in their regulation. The negative and positive effects of light have been documented on gene expression, as the genes *blu* (blue-light-regulated) or *bld* (blue-light-downregulated) were induced or suppressed when *T. atroviride* was exposed to white or blue light [35].

Considering that 6-PP inhibits the growth of plant pathogens [16], and *T. atroviride* produces different compounds regulated by light, we consider that this may influence its antagonism. The antagonistic activity was higher against the pathogens infecting plant roots in *G. graminis*, *R. solani*, *S. sclerotiorum*, and *P. cinnamomi* [27,28,29,30] than it was against pathogens mainly infecting aerial parts of the plant in *A. alternata*, *B. cinerea*, *C. graminicola*, and *F. oxysporum* [25,26,31,32]. These results suggest that *T. atroviride* is a better antagonist in the rhizosphere where light is absent, and the production of 6-PP and other metabolites with antifungal activity is greater, as proposed by Mendoza-Mendoza et al. [16].

In the dark, the growth inhibition was 100% for *P. cinnamomi*, *G. graminis*, and *S. sclerotiorum*, but light weakened this antagonistic capacity. This result is important for assays to select better *Trichoderma* antagonists, which should be conducted in the dark and not in illuminated growth chambers when the main objective is to identify biocontrol isolates for use in plant protection. Furthermore, the strong inhibition of *R. solani* was independent of light, suggesting that the metabolite(s) acting as inhibitors are produced equally well in both light and dark regimes. Although less effective, the results were similar for *B. cinerea*, *C. graminicola*, and *F. oxysporum*. If the compound(s) are the same, they are more efficient in antagonizing Basidiomycetes than Ascomycetes. However, this suggests that *T. atroviride* produces several different compounds that can antagonize other fungi.

Our results provide evidence that should be considered when better antagonists based on *Trichoderma* are being selected. Another interesting issue is how light could affect the performance of *Trichoderma* for different application methods, particularly for foliar application in the field for crop protection.

## 5. Conclusions

Using ethyl acetate as a solvent for extraction of 6-PP from the media was a rapid and highly efficient process, including the direct preliminary concentration of this metabolite. Light regulation of metabolite production was widespread. Several compounds were differently regulated in *Trichoderma,* while 6-PP production was species-dependent. The ability of *T. hamatum*, *T. asperellum*, *T. gamsii*, and *T. viridescens* to produce 6-PP was differentially affected by light in submerged, or on PDA, media. The antagonistic capacity of *T. atroviride* against pathogens of plant roots was greater in the dark than in light, and this response also occurred in other species of *Trichoderma*.

## Figures and Tables

**Figure 1 jof-11-00009-f001:**
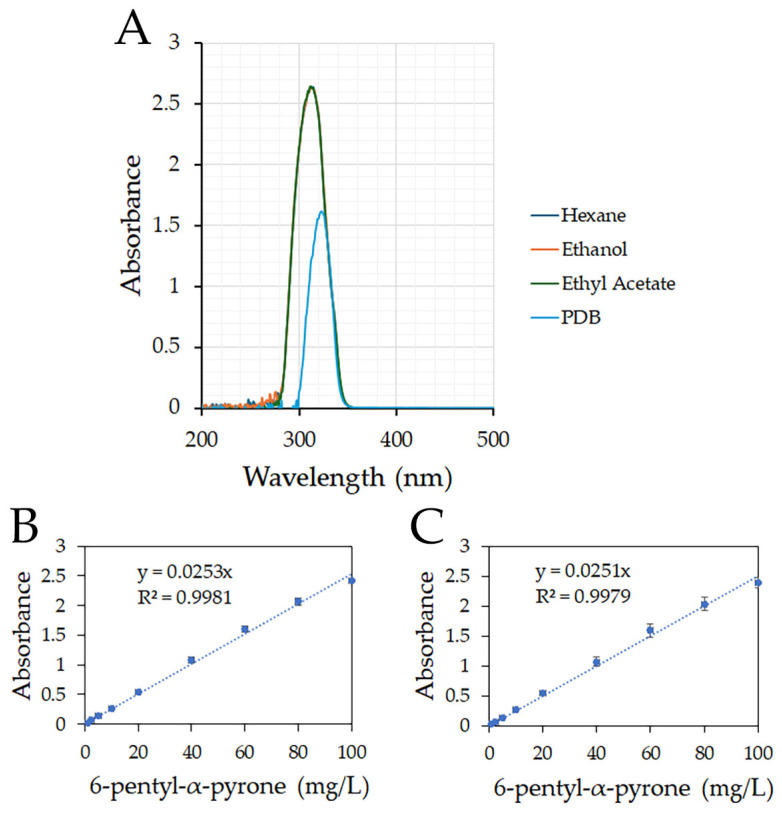
The absorbance of 6-pentyl-α-pyrone in different solvents. (**A**) Absorbance spectrum of 6-PP (100 mg/L) dissolved in *n*-hexane, ethanol, ethyl acetate, and PDB. Absorbance was scanned using a UV-1600PC VWR Spectrophotometer for 200 nm to 500 nm wavelengths. (**B**,**C**) Linear dynamic range. Absorbance was measured at 320 nm using a gradient of concentrations of 6-PP dissolved in ethanol (**B**) and ethyl acetate (**C**).

**Figure 2 jof-11-00009-f002:**
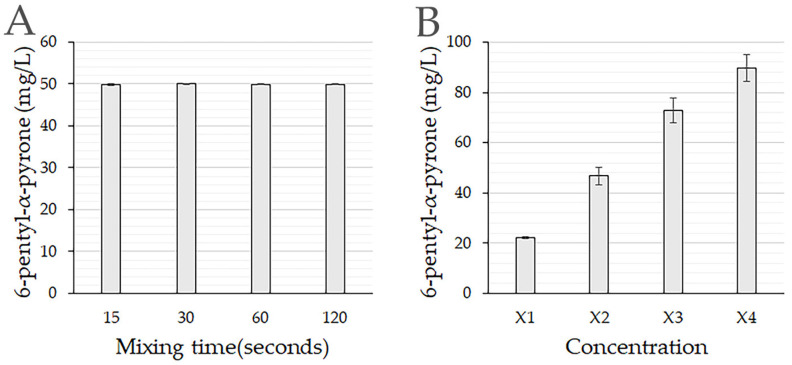
Effect of mixing time and media concentration on extraction of 6-pentyl-α-pyrone. (**A**) Mixing time for recovery of 6-PP (100 mg/L) from the PDB media. The 6-PP dissolved in PDB was mixed with the solvent ethyl acetate, vortexed for the time indicated, and the supernatant recovered by centrifuging. (**B**) Media:Solvent (M:S) ratio for recovery of 6-PP (20 mg/L). Volume relation was 1M:1S (**X1**), 2M:1S (**X2**), 3M:1S (**X3**), and 4M:1S (**X4**). Mixtures were vortexed for 30 s and centrifuged for 3 min at 3000 rpm. The 6-PP was quantified using the equation in Figure 1C.

**Figure 3 jof-11-00009-f003:**
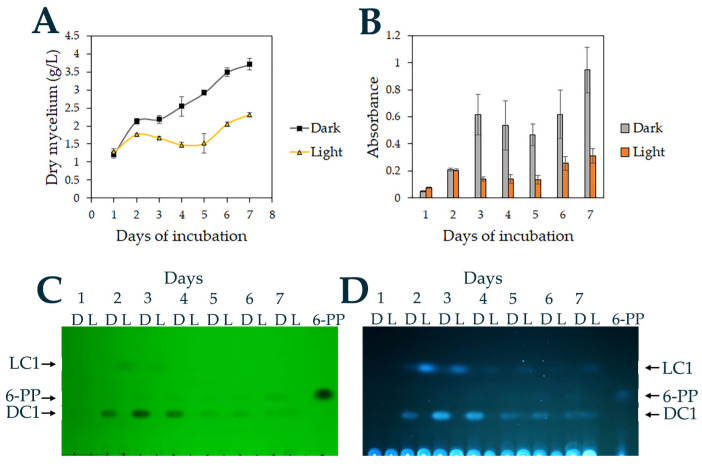
Effect of incubation in the dark or light on *T. atroviride* IMI206040 growth and secretion of organic compounds. (**A**) Dry weight of mycelium after growth in PDB for 7 days at 27 °C. (**B**) Absorbance following ethyl acetate extraction from the *T. atroviride* IMI206040 cultures. (**C**,**D**) Compounds extracted from cultures of *T. atroviride* IMI206040 growing in light or dark for 7 days. The TLCs were exposed to short-wave UV (**C**, 254 nm) or long-wave UV (**D**, 350 nm) to detect the compounds produced in light (LC) or dark (DC).

**Figure 4 jof-11-00009-f004:**
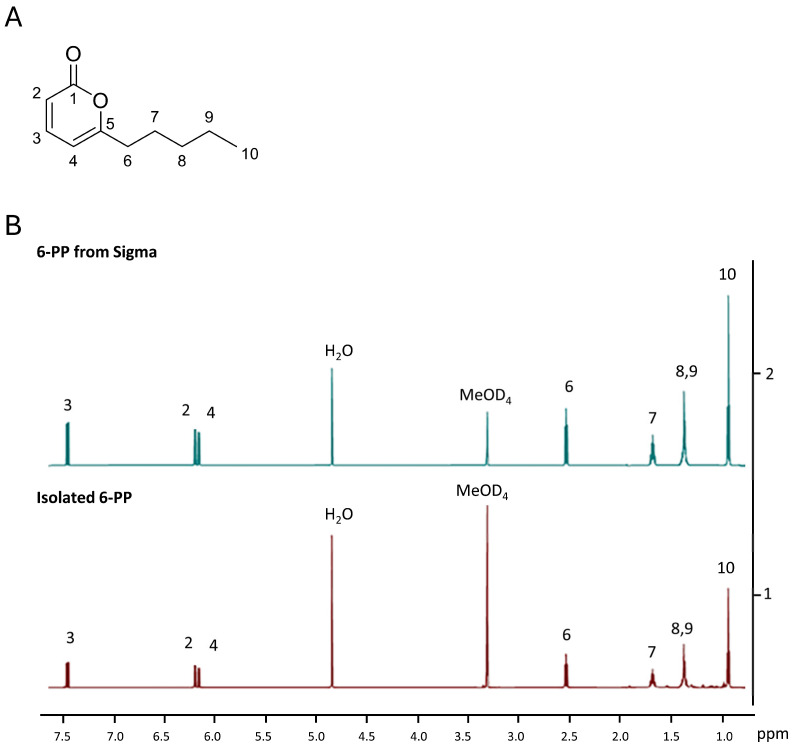
A 6-pentyl-α-pyrone purified from *T. atroviride* IMI206040. (**A**) The Nuclear magnetic resonance of 6-PP purified. (**B**) Stacked NMR spectra of commercially available 6-PP and the 6-PP-like compound isolated from a submerged culture grown in the dark.

**Figure 5 jof-11-00009-f005:**
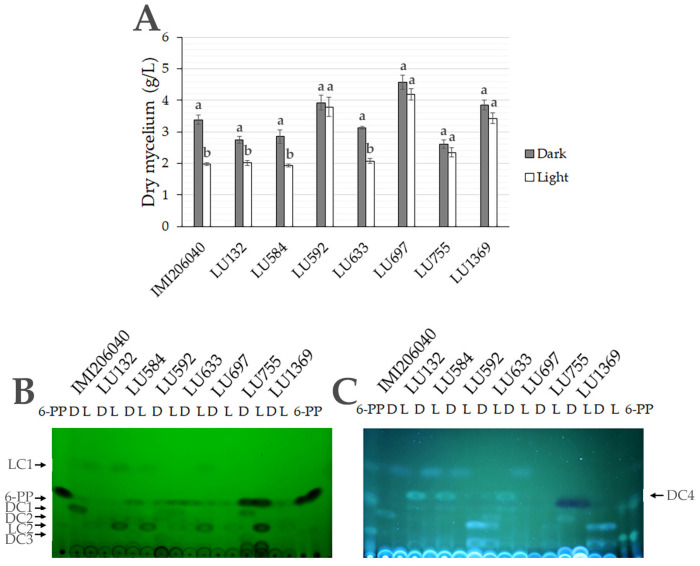
Effect of incubation in the light on growth of *Trichoderma* species/ strains and metabolite production. (**A**) Dry weight of mycelium of the *Trichoderma* strains after growing in PDB for 3 days at 27 °C under light or dark. (**B**,**C**), Metabolite pattern of different strains of *Trichoderma* growing for 3 days at 27 °C in light (L) and dark (D). (**B**) The TLCs exposed to short-wave UV (254 nm) (left) or long-wave UV (350 nm) (right) to detect the compounds produced in light (LC) or dark (DC). The strains used were *T. atroviride* IMI206040, *Trichoderma* sp. *atroviride* B LU132, LU584, and LU633, *T. hamatum* LU592, *T. asperellum* LU697, *T. gamsii* LU755, and *T. viridescens* LU1369. The bars have the (+/−) standard deviation of data generated from three replicates, and different letters over the bars indicate significant differences between dark and light.

**Figure 6 jof-11-00009-f006:**
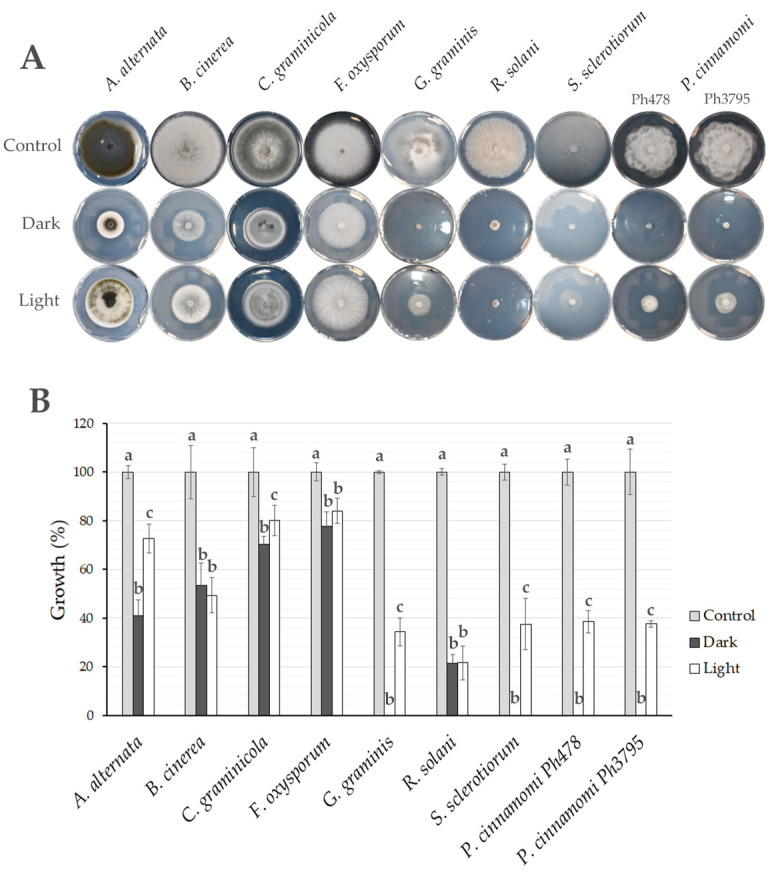
Effect of growth in the dark and light on antagonistic activity of *T. atroviride* IMI206040 as assessed by plant pathogen growth. (**A**) Antagonistic activity on PDA plates. (**B**) Colony growth: colony diameter was measured using ImageJ, and the average diameter of the control was taken as 100 % growth. The vertical bars on each bar are (+/−) standard deviation of data generated from three replicates, and different letters over the bars indicate significant differences between dark and light.

**Figure 7 jof-11-00009-f007:**
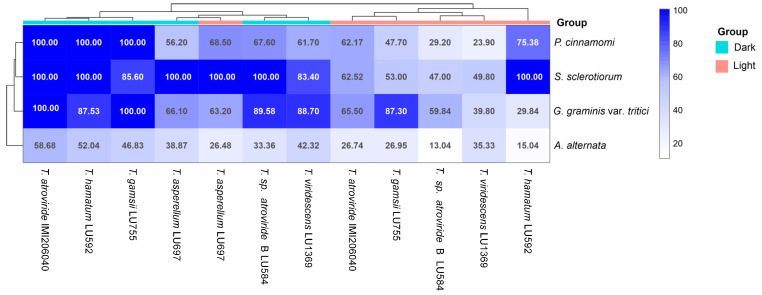
Effect of light on antagonistic activity of several species of *Trichoderma* against four plant pathogens. Antagonistic activity was assayed by growing *Trichoderma* strains on PDA plates covered with a cellophane sheet for 48 h in the dark or light at 27 ° C. After removing the *Trichoderma* colony, the plant pathogens were inoculated, and as a control, fresh PDA was inoculated. Each treatment had three replicates (Appendix A). The heat map matrix shows the average inhibition percentage of two treatments: dark and light.

**Figure 8 jof-11-00009-f008:**
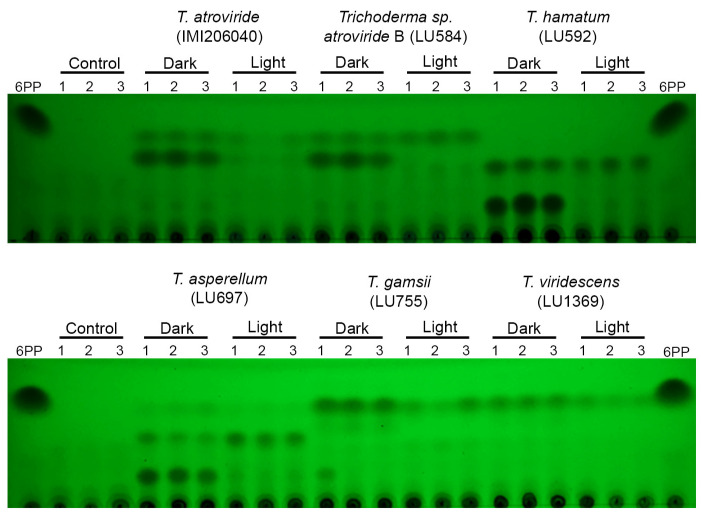
Metabolites secreted by *Trichoderma* spp. grown on PDA in the dark or light. PDA plates holding *Trichoderma* growth for 60 h in the dark or light were treated to isolate the metabolites produced by *Trichoderma* strains and analyzed by TLC. The compounds from three cultures were detected using short-wave UV. As a control, metabolite extraction was performed using fresh PDA, and as a reference, 5 µg 6-PP was used.

## Data Availability

The original contributions presented in this study are included in the article/Appendix A; further inquiries can be directed to the corresponding authors.

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
