# Peer review of "Light Regulates Secreted Metabolite Production and Antagonistic Activity in Trichoderma"

_jof, 2024, doi:10.3390/jof11010009_

Round 1
Reviewer 1 Report
This manuscript had described the important influence of light to growth status, secondary metabolism, especially 6-pp, and their antagonistic activity of Trichoderma.
(1) Figure 3C-D, HPLC spectra could show more abundant information of secreted metabolites than those of TLC, especially 210 nm and 320 nm. Figures 6 and 9 should also replace TLC with HPLC. In fact, detailed comparisons of their MS data should be more meaningful than those of HPLC and TLC.
(2) Figure 3-B, only OD320 should not accurately show the content of 6-pp. First, the standard curve of 6-pp should be determined using HPLC; then, peak area of 6-pp in the EtOAc extract of PDB culture should be calculated; finally, the accurate content of 6-pp could be calculated on the basis of its standard curve.
(3) Figure 4, NMR data should be transferred in the main text, not in the figure title. 1H and 13 C NMR data should also list their corresponding H and C numbers.
(4) Figure 5, besides the NMR comparison, HPLC comparison of 6-pp (sigma) and the EtOAc extract of PDB culture should also be shown.
This manuscript had described the important influence of light to growth status, secondary metabolism, especially 6-pp, and their antagonistic activity of Trichoderma.
(1) Figure 3C-D, HPLC spectra could show more abundant information of secreted metabolites than those of TLC, especially 210 nm and 320 nm. Figures 6 and 9 should also replace TLC with HPLC. In fact, detailed comparisons of their MS data should be more meaningful than those of HPLC and TLC.
(2) Figure 3-B, only OD320 should not accurately show the content of 6-pp. First, the standard curve of 6-pp should be determined using HPLC; then, peak area of 6-pp in the EtOAc extract of PDB culture should be calculated; finally, the accurate content of 6-pp could be calculated on the basis of its standard curve.
(3) Figure 4, NMR data should be transferred in the main text, not in the figure title. 1H and 13 C NMR data should also list their corresponding H and C numbers.
(4) Figure 5, besides the NMR comparison, HPLC comparison of 6-pp (sigma) and the EtOAc extract of PDB culture should also be shown.
Author Response
Major comments
This manuscript had described the important influence of light to growth status, secondary metabolism, especially 6-pp, and their antagonistic activity of Trichoderma.
(1) Figure 3C-D, HPLC spectra could show more abundant information of secreted metabolites than those of TLC, especially 210 nm and 320 nm. Figures 6 and 9 should also replace TLC with HPLC. In fact, detailed comparisons of their MS data should be more meaningful than those of HPLC and TLC.
Response: Thanks for your comments. We agree that HPLC and MS are much more potent metabolite analysis techniques. But by using TLC for metabolite detection we were able to clearly detect metabolite production differences in the dark and light, readily standardize the extraction of metabolites, analyse hundreds of samples, demonstrate the effect of light in Trichoderma species and finally demonstrate the impact on the antagonism of several species of Trichoderma. In our opinion, these results will be valuable for future studies of metabolites in any microorganism, and specifically the impact on understanding Trichoderma biology and the effect of one of the most critical environmental factors regulating one of the most relevant processes in several species of Trichoderma for plant protection.
(2) Figure 3-B, only OD320 should not accurately show the content of 6-pp. First, the standard curve of 6-pp should be determined using HPLC; then, peak area of 6-pp in the EtOAc extract of PDB culture should be calculated; finally, the accurate content of 6-pp could be calculated on the basis of its standard curve.
Response: Thanks for your comments. First, we carefully indicate in our manuscript absorbance or optical density (OD) to describe the time when absorbance increased as a correlation with major metabolite production, but we did not indicate the concentration of 6-PP (see lines 244-250), which would be inappropriate because many compounds were suspended in the solvent when media of the Trichoderma cultures were used for extraction. Second, we know that HPLC can be used to make a standard curve, but absorbance was useful to standardise the extraction using the 6-PP from SIGMA and PDB. Then, TLC helped demonstrate that the extraction with ethyl acetate works very well, identifying 6-PP as one of the compounds produced by Trichoderma.
(3) Figure 4, NMR data should be transferred in the main text, not in the figure title. 1H and 13 C NMR data should also list their corresponding H and C numbers.
Response: Thanks for your suggestion. Data were transferred to the main text.
(4) Figure 5, besides the NMR comparison, HPLC comparison of 6-pp (sigma) and the EtOAc extract of PDB culture should also be shown.
Response: Thanks for your comments. We focused on identifying that the compound with the same mobility as 6-PP was the same to validate our approach. Instead of knowing how much 6-PP Trichoderma is producing, we were interested in demonstrating if there was a correlation between metabolite production and antagonistic activity.
Response: We appreciate your effort in reviewing and your feedback.
Reviewer 2 Report
The paper entitled: “Light regulates secreted metabolite production and antagonistic activity in Trichoderma”, describes the role of light as a regulatory element in the production of secondary metabolites, in different species of Trichoderma, especially in the production of the compound 6-pentyl-alpha-pyrone (6PP).
The authors also study an efficient extraction method for 6PP using different solvents and different extractive methods that have allowed them to purify the compound from submerged cultures using a conventional medium, such as PDB.
And finally, the authors carry out experiments under light and dark conditions, which confirm that the Trichoderma strains used in this study, in general, maintain good antagonistic activity against different plant pathogenic microorganisms when kept in dark conditions.
The work is good, it is clear that the authors have taken care of a number of details, but it would be necessary to improve the presentation in the following sections:
The work is good, it is clear that the authors have taken care of a number of details, the introduction is well documented, it is brief and sets out very clearly what the objectives of the work are.
Points to review in the following sections (minor):
Materials and methods
Section 2.1, on line 91, where it says: "The liquid medium", should say "The liquid medium cultures"
Section 2.7, between lines 159 and 162 there are two sentences that seem repetitive. Perhaps the authors could say: "5 g of sodium sulphate Na2SO4 were added, the mixture was allowed to stand for 10 min before being filtered through a cotton plug and the organic phase was concentrated in vacuo"
Section 2.9, line 128, please, add the diameter in cm of the Petri dishes
In section 3.3, add at least the range, in mg/L, of the production observed in graph 3B, in the presence of light and in darkness, although a certain concentration can be intuited from the absorbance measurement, the data must be reflected volumetrically as the quantity in mg/L of 6PP or as the quantity in mg/L of 6PP per mg of dry weight (specific production), if not in the text, then in the footnote of figure 3.
The rest of the work is in agreement with the initial approaches.
The discussion is good
The authors' conclusions should add that ethyl acetate is a solvent that is routinely used for the extraction of a good number of compounds, and is therefore a universal solvent that in this case has proven to be equally effective for the recovery of 6PP.
Finally, authors should write complete references for the work, as they are written in other articles published in this journal.
Thanks.
Author Response
Major comments:
The paper entitled: “Light regulates secreted metabolite production and antagonistic activity in Trichoderma”, describes the role of light as a regulatory element in the production of secondary metabolites, in different species of Trichoderma, especially in the production of the compound 6-pentyl-alpha-pyrone (6PP).
The authors also study an efficient extraction method for 6PP using different solvents and different extractive methods that have allowed them to purify the compound from submerged cultures using a conventional medium, such as PDB.
And finally, the authors carry out experiments under light and dark conditions, which confirm that the Trichoderma strains used in this study, in general, maintain good antagonistic activity against different plant pathogenic microorganisms when kept in dark conditions.
The work is good, it is clear that the authors have taken care of a number of details, but it would be necessary to improve the presentation in the following sections:
The work is good, it is clear that the authors have taken care of a number of details, the introduction is well documented, it is brief and sets out very clearly what the objectives of the work are.
Response: Thank you for your feedback; we are responding to all comments.
Detail comments:
Points to review in the following sections (minor):
Materials and methods
Section 2.1, on line 91, where it says: "The liquid medium", should say "The liquid medium cultures"
Response: Thank you for your suggestion. We added cultures to line 91.
Section 2.7, between lines 159 and 162 there are two sentences that seem repetitive. Perhaps the authors could say: "5 g of sodium sulphate Na2SO4 were added, the mixture was allowed to stand for 10 min before being filtered through a cotton plug and the organic phase was concentrated in vacuo"
Response: Thank you for your observation. We deleted the following duplicated sentence (line 161-162):
The organic phase was dried (Na2SO4), filtered through a cotton plug, and concentrated in vacuo.
Section 2.9, line 128, please, add the diameter in cm of the Petri dishes
Response: Thank you for your suggestion. The diameter of Petri dishes was indicated in parenthesis (9 cm).
In section 3.3, add at least the range, in mg/L, of the production observed in graph 3B, in the presence of light and in darkness, although a certain concentration can be intuited from the absorbance measurement, the data must be reflected volumetrically as the quantity in mg/L of 6PP or as the quantity in mg/L of 6PP per mg of dry weight (specific production), if not in the text, then in the footnote of figure 3.
Response: Thank you for your comment. We can't indicate the 6-PP concentration because several compounds were extracted with very similar absorbance (for more details, see Figure 5S). Instead, we decided to use optical density at 320, which we found to be linear with a 6-PP concentration. We followed absorbance only to standardize the extraction of secreted metabolites of Trichoderma. It is very important to indicate that the absorbance is not specific for 6-PP when we work with a mixture of compounds extracted from Trichoderma media, such as we showed by TLC.
The rest of the work is in agreement with the initial approaches.
Response: Thank you for your nice comment.
The discussion is good
Response: Thank you. We appreciate your comment.
The authors' conclusions should add that ethyl acetate is a solvent that is routinely used for the extraction of a good number of compounds, and is therefore a universal solvent that in this case has proven to be equally effective for the recovery of 6PP.
Response: Thank you for your comment. Ethyl acetate is routinely used, and in fact, it has been used to extract 6-PP from Trichoderma in several papers, but we want to highlight the improvements in our work. While the previous protocols required a significant amount of solvent and time for extraction, we demonstrated that using small volumes of ethyl acetate is highly efficient in seconds. Furthermore, we demonstrate that Trichoderma produced the maximum amount of 6-PP in three days while previous works used one or more weeks of fermentation to extract and analyse 6-PP.
Finally, authors should write complete references for the work, as they are written in other articles published in this journal.
Response: Thank you so much for your comment, and the references are complete.
Thanks.
Response: We appreciate your effort in reviewing and your feedback.
Reviewer 3 Report
The authors described in the manuscript intituled "Light regulates secreted metabolite production and antagonistic activity in Trichoderma": a) standardization of the method for extraction of secondary metabolites using different organic solvents, b) investigation of the impact of light and dark on the metabolites production, and c) they presented how the light impacts the antagonistic activity of Trichoderma atroviride against several fungal plant pathogens. The authors also identified and purified 6-PP from extracts of T. atroviride, by NMR and flash chromatography, respectively. The work addresses a very interesting research area. I believe that this study can be a valuable source of information for other researchers in the field. Minor revisions are required. The English could be improved all over the text.
I found the manuscript informative and complete. Some comments were made within the text, which is attached.

Author Response
Major comments:
The authors described in the manuscript intituled "Light regulates secreted metabolite production and antagonistic activity in Trichoderma": a) standardization of the method for extraction of secondary metabolites using different organic solvents, b) investigation of the impact of light and dark on the metabolites production, and c) they presented how the light impacts the antagonistic activity of Trichoderma atroviride against several fungal plant pathogens. The authors also identified and purified 6-PP from extracts of T. atroviride, by NMR and flash chromatography, respectively. The work addresses a very interesting research area. I believe that this study can be a valuable source of information for other researchers in the field. Minor revisions are required. The English could be improved all over the text.
Response: Thank you so much for your comments. The English has been improved.
Detail comments:
I found the manuscript informative and complete. Some comments were made within the text, which is attached
Authors should write these abbreviations at least once: PDB and PDA.
Response: Thank you for your suggestion. We indicate the complete name potato-dextrose-broth for PDB and potato-dextrose-agar for PDA in the abstract, and we have also described such abbreviations at the first mention in the manuscript.
Confuse! Re-write
Equation of the line? Or Calibration curve?
Response: Thank you for your suggestion. To clarify, we deleted the sentence because the standard/calibration curve generated implied that the data were analyzed by linear regression, which sounds redundant.
Uniform 6-pp
Response: Thank you for your observation. We changed 6-pp to 6-PP.
(60 F245, MERCK)
Response: Thank you for your observation. We changed (60 F245, MERCK) to (60 F245, MERCK).
Improve the equipment description. Maybe it would be better to use 9.4 Tesla.NMR Spectroscopy
Response: Thank you for your suggestion.
Do the authors have some ideia about this compound LC1?
Response: Thank you for your comment. We have no idea about the identity of LC1. To validate our approach we focused only on 6-PP identification.
The authors could highlight, in a summarized manner, the importance of this work, for example, in agricultural or other applications.
Response: Thank you for your suggestion. Our results provide beneficial information for producers on how to select better antagonists of Trichoderma and for agricultural professionals on how light can impact Trichoderma performance.
At the end of discussion we added:
Our results provide evidence that should be considered when better antagonists based on Trichoderma are being selected. Another interesting issue is how light could affect the performance of Trichoderma for different application methods, particularly for foliar application in the field for crop protection.
Response: Thank you for your suggestions. All suggestions in the manuscript were carefully considered. We appreciate your effort in reviewing and your feedback.
Reviewer 4 Report
The work is solid and offers a clear result and several figures to follow the results. However it is not clear from the biological point of view, how many samples were used, how many treatments were done, the statistical methods are absent, and the biological context and the socioeconomical relevance are not introduced neither discussed. The work is only focused as chemical procedure, however as it is not a novel molecule the authors tried with a biological journal. As it is looks more the target of phytochemistry or molecules, since it lacks of mycological/biological relevance in the focus. I suggest to the authors to try to do an effort in this way, I will not offer the detailed way of how to focus more into mycological/biological point of view. Just few comments which are not clear to me or specific suggestions or criticisms, which are detailed below.
The introduction is clear, giving understanding of the state of the art and pointing to the main objective which is standardize the method to analyze the 6-PP and study the role of light in the protective effect of Trichoderma over plants.
The methods are clear from the point of view of the chemical procedures assayed, however are biologically in the shadow. It is not clear how many cultures of each species were used, how the replicates were done, how many samples were used for each experiment. Authors refer to number of replicates but did not refer to how many strains were used for each purpose. And the statistical methods are not described.
How many samples of each Trichoderma species and pathogen species were used? Each Trichoderma species were assayed against each plant pathogen species? How many replicates per treatment were used?
The authors collect several data and include many figures, even more considering the supplementary figures being not that easy to follow the results section. However, the authors could help to clarify the results modifying the redaction and specially explaining better which are the treatments and which species/strains are being used in which treatment. Most of this should be done in MM section, but it will help to follow results section.
The discussion is omitting the biological/mycological point of view. It lacks of of context, biological context, economical context, which can be important for mycologist, pytopathologists, biologists and plant producers which are potentially readers of the journal. For example, the higher inhibition of growth of plant pathogens is in the dark, and the Thricoderma species are root inhabitants with protective action to the plants. The authors have the opportunity to discuss how it can be related to the adaption to the darkness environment of the roots. Discuss about other molecules which have better production in dark, how the circadian cycles affect the biosynthetic pathways…
L 33 With “Many” you refer to “Many Trichoderma species”? If yes, write it to favor understanding.
L 41 I suggest to change pathogens for plant pathogens.
L 84 What “type B” means? Describe it, or at least add a reference.
L 87-89 Did you used any antibiotic to avoid bacterial growth?
L 94-97 Don`t use commas to separate above-ground, root or both (use other separator; or.) since comma are being used to separate species within plant parts.
L 106 Data were analyzed in Excel? Did you mean by using the function maximum? It is almost not an analysis as much is a description of a result. If you used some specific formula or complement describe it here.
L 122 Sometime the reader is not reading the full text or full sections. Add in the subsection how it will be measured (by UV-VIS?).
132 Each strain per separate or you did a pool of strains?
L 135 three replicates of how many treatments? Did you 3 replicates for each strain?
L 135 When you filtered 25 mL, you maintain the other 100 mL of culture shaking, you replace it with fresh media, or the sampled flasks did not come back to the shaker? The three replicates come from different flasks, or in fact it was pseudoreplicates from the same flask? It is not clear to me. To my opinion you should modify this sentence to clarify.
L 176-188 Add some reference.
Subsection 3.1: I understand that this result is measuring from commercial 6-PP. I suggest to specify if it comes from commercial or Trichoderma.
L 217-218 “between 6-PP concentration and absorbance for both ethanol (Fig. 1B) and ethyl acetate (Fig. 1C).” I needed to read this sentence twice, I understand that you are measuring only 6-PP in different solvents, first time I read it I understood that ethanol and ethyl acetate provoked an interference with 6-PP. I suggest to rewrite it to make it clearer.
L 226 From the MM section it is not clear to me how many treatments you used and which are the treatments (different solvents, different concentrations, different strains). You should clarify treatments; a table could help.
L 280-281 Only for this strain IMI206040? You have several strains, only tested in one, why? Or you tested with negative result? It is important to point if it is an effect which is maintained on the species or is just in the strain.
L 306 here is the first time that I see which species/strains were used from the the section 2.1.
Clarify in the MM subsections and previous results subsection which species/strains were used.
Did you used the different strains/species only for the light/pathogens assay? Specify it even to confirm or to explain.
L 381 Do you have several strains of T. atroviride at least three are “type B”, why if only the LU584 showed the result you feel the need to point that is type B? Explain from the beginning (ideal from introduction) at least from MM section what type be is and why is relevant.
L 460-461 Rewrite it, the capacity to “kill” the pathogen was effective, doesn´t reduced the antagonistic capacity, it was 100% in both cases (dark ad light). Your indicator was not good, or the spices used to demonstrate the activity. Even more if you choose only pathogens which have the desired effect it would not be effect of the molecule (6-PP) but also would be responsible for the selection bias.
L 460-461 Rewrite it: even if the inhibition in dark was of 100% we doesn’t found differences in dark/light antagonistic capacity against these planta pathogens since Trichoderma sowed that high virulence against them which inhibited their growth even in light.
Author Response
Major comments:
The work is solid and offers a clear result and several figures to follow the results. However it is not clear from the biological point of view, how many samples were used, how many treatments were done, the statistical methods are absent, and the biological context and the socioeconomical relevance are not introduced neither discussed. The work is only focused as chemical procedure, however as it is not a novel molecule the authors tried with a biological journal. As it is looks more the target of phytochemistry or molecules, since it lacks of mycological/biological relevance in the focus. I suggest to the authors to try to do an effort in this way, I will not offer the detailed way of how to focus more into mycological/biological point of view. Just few comments which are not clear to me or specific suggestions or criticisms, which are detailed below.
The introduction is clear, giving understanding of the state of the art and pointing to the main objective which is standardize the method to analyze the 6-PP and study the role of light in the protective effect of Trichoderma over plants.
The methods are clear from the point of view of the chemical procedures assayed, however are biologically in the shadow. It is not clear how many cultures of each species were used, how the replicates were done, how many samples were used for each experiment. Authors refer to number of replicates but did not refer to how many strains were used for each purpose. And the statistical methods are not described.
Response: Thanks for your comments. We started the work with chemical procedures as they were necessary to standardize the extraction and analysis of metabolites. However, regulating metabolite production impacts one of the most crucial characteristics of the genus Trichoderma in plant protection, as we demonstrated via the antagonistic activity against nine different plant pathogens. Numbers for cultures, strains, species, replicates, etc., are given in the methods and the figure legends.
How many samples of each Trichoderma species and pathogen species were used? Each Trichoderma species were assayed against each plant pathogen species? How many replicates per treatment were used?
Response: Thank you for your comments. The number of samples of each Trichoderma species and pathogen is indicated in the methods and figure legends.
Figure 7. Effect of growth in the dark and light on antagonistic activity of T. atroviride IMI206040 as assessed by plant pathogen growth. A, Antagonistic activity on PDA plates. B, Colony growth: colony diameter was measured using ImageJ and the diameter average of the control was taken as 100 % growth. The vertical bars on each bar are (+/-) standard deviation of data generated from three replicates.
The authors collect several data and include many figures, even more considering the supplementary figures being not that easy to follow the results section. However, the authors could help to clarify the results modifying the redaction and specially explaining better which are the treatments and which species/strains are being used in which treatment. Most of this should be done in MM section, but it will help to follow results section.
Response: Thank you for your comments. We have included information that improves comprehension of the methods, but it is difficult to provide the information for non-specific questions.
The discussion is omitting the biological/mycological point of view. It lacks of of context, biological context, economical context, which can be important for mycologist, pytopathologists, biologists and plant producers which are potentially readers of the journal. For example, the higher inhibition of growth of plant pathogens is in the dark, and the Thricoderma species are root inhabitants with protective action to the plants. The authors have the opportunity to discuss how it can be related to the adaption to the darkness environment of the roots. Discuss about other molecules which have better production in dark, how the circadian cycles affect the biosynthetic pathways…
Response: Thank you for your comments. We discussed our results and how light has been affecting the selection of better antagonistic strains for crop protection.
Line 474-477, “In the dark, the growth inhibition was 100% for P. cinnamomic, G. graminis, and S. sclerotiorum, but light weakened this antagonistic capacity. This result is important for assays to select better Trichoderma antagonists, which should be conducted in the dark and not in illuminated growth chambers when the main goal is plant protection.”
Detail comments:
L 33 With “Many” you refer to “Many Trichoderma species”? If yes, write it to favor understanding.
Response: Thank you for your suggestion, we change “Many” for “Many Trichoderma species”.
L 41 I suggest to change pathogens for plant pathogens.
Response: Thank you for your suggestion. We use plant pathogen instead of pathogen.
L 84 What “type B” means? Describe it, or at least add a reference.
Response: Type B of T. atroviride is explained in Braithwaite et al ( 2016), and we used the same name as the authors indicated (Trichoderma sp. atroviride B).
L 87-89 Did you used any antibiotic to avoid bacterial growth?
Response: Thanks for your comment. Yes, we used chloramphenicol to avoid bacterial growth and to ensure that we were only growing the indicated fungi.
L 94-97 Don`t use commas to separate above-ground, root or both (use other separator; or.) since comma are being used to separate species within plant parts.
Response: Thanks for your comment.
L 106 Data were analyzed in Excel? Did you mean by using the function maximum? It is almost not an analysis as much is a description of a result. If you used some specific formula or complement describe it here.
Response: Thank you for your comment. We improved the information: “A spectrum scan was carried out from 200 to 700 nm using a UV-1600PC spectrophotometer (VWR) to identify the maximum absorbance peak at different wavelengths.”
The spectrophotometer saves data in files compatible with Excel, so we analysed all the data (thousands) much faster, created nice graphs, and identified the maximum peak.
L 122 Sometime the reader is not reading the full text or full sections. Add in the subsection how it will be measured (by UV-VIS?).
Response: Thank you for your comment. UV-VIS implies that we analysed the absorbance to detect molecules in dissolution using different wavelengths from ultraviolet (UV) to the visible (VIS) spectrum, making a scan from 200 to 700 nm.
132 Each strain per separate or you did a pool of strains?
Response: Thanks so much for your observation. We improved this section, including the indication of strains used in each experiment.
Trichoderma atroviride IMI206040 (1X106 conidia per mL) was added to 125mL flasks containing 25 mL of PDB and incubated in the constant dark and constant white light or under photoperiods (12h dark:12h light) at 27 °C for 7 days on a shaker at 120 rpm. Every 24 hours, three replicates of 25 mL of the cultures were filtered using funnels through a miracloth membrane (Merck Millipore). The filtrates were placed into plastic falcon tubes for metabolite analysis, and the miracloth containing the mycelia was dried at 80 °C for 24 h to determine mycelium dry weight. The metabolites were extracted by mixing 3 mL of filtrate with 3 mL of ethyl acetate, vortexed for 1 min at maximum speed (3,400 rpm), and then centrifuged for 3 min at 3,000 rpm. The resultant SN was used to analyse the absorbance spectrum using a UV-1600PC spectrophotometer (VWR) and thin layer chromatography (TLC) was performed by mixing 2 mL of each replicate to get 6 mL and 1.5 mL of ethyl acetate.
Metabolites from different Trichoderma strains (LU132, LU584, LU633, LU592, LU697, LU755, LU1369) were compared by inoculating 1X106 conidia per mL in flasks with 25 mL of PDB incubated in the dark or light for 3 days at 27 °C. The metabolites were analysed in triplicate, as indicated above.
L 135 three replicates of how many treatments? Did you 3 replicates for each strain?
Response: Thank you for your questions. We always carried out at least three biological replicates in each experiment for each strain.
L 135 When you filtered 25 mL, you maintain the other 100 mL of culture shaking, you replace it with fresh media, or the sampled flasks did not come back to the shaker? The three replicates come from different flasks, or in fact it was pseudoreplicates from the same flask? It is not clear to me. To my opinion you should modify this sentence to clarify.
Response: Thanks for your concerns. We used flasks with a capacity of 125 mL, and each flask contained 25 mL PDB. The three replicates came from different flasks: it is correct.
L 176-188 Add some reference.
Response: Thanks for your suggestion. We did not use any reference for this method.
Subsection 3.1: I understand that this result is measuring from commercial 6-PP. I suggest to specify if it comes from commercial or Trichoderma.
Response: Thanks for your comment. We indicated in method 2.2.
2.2. UV-VIS analysis of 6-pentyl-α-pyrone in different solvents
6-PP (Sigma-Aldrich, St. Louis, MO, USA) was dissolved in solvents with different relative polarities (RP): hexane (RP=0.006, MERK, Germany), ethyl acetate (RP=0.228, HPLC grade, Fisher Chemical), ethanol (RP=0.654, absolute AR grade, LAB Supply) and PDB dissolved in water (RP=1)
L 217-218 “between 6-PP concentration and absorbance for both ethanol (Fig. 1B) and ethyl acetate (Fig. 1C).” I needed to read this sentence twice, I understand that you are measuring only 6-PP in different solvents, first time I read it I understood that ethanol and ethyl acetate provoked an interference with 6-PP. I suggest to rewrite it to make it clearer.
Response: Thanks for your comment.
L 226 From the MM section it is not clear to me how many treatments you used and which are the treatments (different solvents, different concentrations, different strains). You should clarify treatments; a table could help.
Response: Thanks for your comment. We indicate the treatment as mixing time and the sentence read as follow:
For all mixing times, 6-PP was rapidly detected in the supernatant, with the shortest mixing time (15 seconds) was sufficient to recover all the 6-PP dissolved in PDB (Fig. 2A). This indicates that ethyl acetate is highly efficient in dissolving this compound and facilitating 6-PP recovery from the media. With this solvent's very low solubility in water, the supernatant was separated after 10 minutes without any additional agitation.
L 280-281 Only for this strain IMI206040? You have several strains, only tested in one, why? Or you tested with negative result? It is important to point if it is an effect which is maintained on the species or is just in the strain.
Response: Thanks for your comment. We first standardised the metabolite extraction using T. atroviride IMI206040 and then used this approach to explore the regulation of metabolite production in other Trichoderma strains and species. We found some differences that we described in the main manuscript, but the extraction approach always works perfectly.
L 306 here is the first time that I see which species/strains were used from the the section 2.1.
Response: Thanks for your comment. Our strategy first contemplated standardizing the extraction using one of the most studied mycoparasites, such as T. atroviride (IMI206040). Next, we validated the approach, identifying the compound with the same Rf using 6-PP from Sigma as a reference. Then, we expanded the potential of our approach using several strains representing several species of Trichoderma. For that reason, several species/strains are found later in the document.
Clarify in the MM subsections and previous results subsection which species/strains were used.
Response: Thanks for your comment. We clarify which species/strains were used.
2.1. Trichoderma strains and plant pathogens
The Trichoderma strains used in this work were Trichoderma atroviride IMI206040 (ATCC 204676), Trichoderma sp. atroviride B LU132, Trichoderma sp. atroviride B LU584, and Trichoderma sp. atroviride B LU633, T. hamatum LU592, T. asperellum LU697, T. gamsii LU755, and T. viridescens LU1369. The strains with LU codes were New Zealand isolates obtained from the Lincoln University Trichoderma collection.
2.5. Extraction of metabolites from liquid media
Trichoderma atroviride IMI206040 (1X106 conidia per mL) was added to 125mL flasks containing 25 mL of PDB and incubated in the constant dark and constant white light or under photoperiods (12h dark:12h light) at 27 °C for 7 days on a shaker at 120 rpm. Every 24 hours, three replicates of 25 mL of the cultures were filtered using funnels through a miracloth membrane (Merck Millipore). The filtrates were placed into plastic falcon tubes for metabolite analysis, and the miracloth containing the mycelia was dried at 80 °C for 24 h to determine mycelium dry weight. The metabolites were extracted by mixing 3 mL of filtrate with 3 mL of ethyl acetate, vortexed for 1 min at maximum speed (3,400 rpm), and then centrifuged for 3 min at 3,000 rpm. The resultant SN was used to analyse the absorbance spectrum using a UV-1600PC spectrophotometer (VWR) and thin layer chromatography (TLC) was performed by mixing 2 mL of each replicate to get 6 mL and 1.5 mL of ethyl acetate.
Metabolites from different Trichoderma strains (LU132, LU584, LU633, LU592, LU697, LU755, LU1369) were compared by inoculating 1X106 conidia per mL in flasks with 25 mL of PDB incubated in the dark or light for 3 days at 27 °C. The metabolites were analysed in triplicate, as indicated above.
2.9. Extraction of metabolites from PDA
Fresh PDA plates covered with a cellophane sheet were inoculated in triplicate with Trichoderma strains (IMI206040, LU584, LU592, LU697, LU755 and LU1369) using mycelia discs 5 mm in diameter cut from the colony margin.
Did you used the different strains/species only for the light/pathogens assay? Specify it even to confirm or to explain.
Response: Thank you for question. We used different strains and species to analyse the effect of light on metabolite production and antagonistic activity against different plant pathogens.
L 381 Do you have several strains of T. atroviride at least three are “type B”, why if only the LU584 showed the result you feel the need to point that is type B? Explain from the beginning (ideal from introduction) at least from MM section what type be is and why is relevant.
Response: Thank you for your comment.
Braithwaite et al. 2016 reported the diversity of Trichoderma in New Zealand; the authors identified that Trichoderma sp. atroviride B represented a new type of Trichoderma atroviride. This new type Trichoderma sp. atroviride, is a species present only in the South hemisphere. We have identified multiple strains from this species and currently reported van Zijll de Jong, E, et al 2023. Journal of Fungi (Basel). 9 (2), 238
L 460-461 Rewrite it, the capacity to “kill” the pathogen was effective, doesn´t reduced the antagonistic capacity, it was 100% in both cases (dark ad light). Your indicator was not good, or the spices used to demonstrate the activity. Even more if you choose only pathogens which have the desired effect it would not be effect of the molecule (6-PP) but also would be responsible for the selection bias.
Response: Thanks for your comment, but it is not clear what your suggestion is.
Round 2
Reviewer 1 Report
My cocerns had been responded.
No more comments.
Author Response
Thanks so much, no response is required here.